# Treatment Optimization in Linac-Based SBRT for Localized Prostate Cancer: A Single-Arc versus Dual-Arc Plan Comparison

**DOI:** 10.3390/cancers16010013

**Published:** 2023-12-19

**Authors:** Denis Panizza, Valeria Faccenda, Stefano Arcangeli, Elena De Ponti

**Affiliations:** 1Medical Physics Department, Fondazione IRCCS San Gerardo dei Tintori, 20900 Monza, Italy; valeria.faccenda@irccs-sangerardo.it (V.F.); elena.deponti@irccs-sangerardo.it (E.D.P.); 2School of Medicine and Surgery, University of Milan Bicocca, 20126 Milan, Italy; stefano.arcangeli@unimib.it; 3Radiation Oncology Department, Fondazione IRCCS San Gerardo dei Tintori, 20900 Monza, Italy

**Keywords:** prostate cancer, stereotactic body radiation therapy, extreme hypofractionation, linac, volumetric arc therapy

## Abstract

**Simple Summary:**

Stereotactic body radiation therapy (SBRT) can be safely delivered for the treatment of localized prostate cancer. High-dose-per-fraction treatments require precise and rapid delivery of the radiation dose to the tumor, and any potential margin expansion must be balanced against the risk of increased toxicity. Since significant intrafraction prostate displacement can have a relevant impact on an extreme-hypofractionated regimen, real-time monitoring has emerged as a strategy to avoid the excessive exposure of healthy tissue or insufficient target coverage, but it requires continuous investment of resources and time. In real-world clinical practice, the utilization of organ motion management devices is not always feasible for all patients. Thus, our strategy has often been to mitigate the impact of intrafraction motion by shortening the delivery time without compromising the quality of the treatment. This was achieved through optimal patient preparation, flattening filter-free (FFF) beams, adequate margins, and a reduction in the number of arcs in the VMAT technique.

**Abstract:**

This study aimed to comprehensively present data on treatment optimization in linac-based SBRT for localized prostate cancer at a single institution. Moreover, the dosimetric quality and treatment efficiency of single-arc (SA) versus dual-arc (DA) VMAT planning and delivery approaches were compared. Re-optimization was performed on twenty low-to-intermediate-risk- (36.25 Gy in 5 fractions) and twenty high-risk (42.7 Gy in 7 fractions) prostate plans initially administered with the DA FFF-VMAT technique in 2021. An SA approach was adopted, incorporating new optimization parameters based on increased planning and clinical experience. Analysis included target coverage, organ-at-risk (OAR) sparing, treatment delivery time, and the pre-treatment verification’s gamma analysis-passing ratio. The SA optimization technique has consistently produced superior plans. Rectum and bladder mean doses were significantly reduced, and comparable target coverage and homogeneity were achieved in order to maintain a urethra protection strategy. The mean SA treatment delivery time was reduced by 22%; the mean monitor units increased due to higher plan complexity; and dose measurements demonstrated optimal agreement with calculations. The substantial reduction in treatment delivery time decreased the probability of prostate motion beyond the applied margins, suggesting potential decrease in treatment-related toxicity and improved target coverage in prostate SBRT. Further investigations are warranted to assess the long-term clinical outcomes.

## 1. Introduction

The optimal management of prostate cancer has remained a subject of controversy despite recent advances in early detection and treatment of localized forms. Stereotactic body radiotherapy (SBRT) is a highly conformal extreme-hypofractionated radiation therapy (RT) technique that significantly reduces the total duration of treatment [1,2,3]. The findings of available studies convincingly demonstrate that SBRT can be safely delivered with excellent outcomes in patients with localized prostate cancer [4,5,6,7,8,9,10,11,12,13]. Nonetheless, due to the higher doses per fraction and fewer fractions compared to conventional treatment, SBRT requires more precise dose gradients with tighter margins. Since several studies [14,15,16,17,18,19] have identified substantial and unpredictable motion of the prostate during the beams’ delivery, significant intrafractional displacement can have a relevant impact on treatment, potentially leading to either excessive exposure of healthy tissue or insufficient target coverage [20,21,22,23,24,25,26]. 

While slight target displacements during normo-fractionated treatments are generally tolerable because they occur randomly and average out during the treatment course, their impact becomes more critical in SBRT due to potential long-term effects and reduced statistical averaging of errors. Moreover, in high-dose-per-fraction treatments, it is important to balance any potential margins’ expansion against the risk of increased toxicity, especially if current practices are yielding favorable biochemical control and side effects [27]. Therefore, intrafraction motion management and real-time monitoring have emerged as strategies to ensure accurate radiation delivery and minimize adverse effects but with considerable and continuous investment of resources and time. 

In real-world clinical practice, where reducing the patient waiting list is advantageous and desirable, the implementation and utilization of organ motion management devices is not always feasible for all patients. Usually, for those undergoing non-dose-escalated prostate SBRT, the choice has often been to mitigate the impact of organ motion by optimizing the treatment process. This was achieved using various strategies aimed at shortening SBRT delivery time without compromising the quality of the treatment. These included optimal bowel and bladder preparation, 10 MV flattening filter-free (FFF) beams (2000 MU/min) [28], and the widely used 5 mm isotropic margins for the planning target volume (PTV), with the exception of 3 mm towards the rectum [11,29,30]. Recently, based on the experience gained over past years, the strategy also involved a reduction in the number of arcs used in the volumetric modulated arc therapy (VMAT) technique.

This study aimed to comprehensively present data on the treatment optimization of linac-based SBRT for localized prostate cancer at a single institution. Moreover, the dosimetric quality and treatment efficiency of single-arc (SA) versus dual-arc (DA) VMAT planning and delivery approaches were evaluated and compared. 

## 2. Materials and Methods

Twenty low-to-intermediate-risk and twenty high-risk prostate cancer volumetric intensity-modulated arc therapy (VMAT) treatment plans delivered during the year 2021 on a VersaHD linear accelerator (Elekta AB, Stockholm, Sweden) were selected. Prescribed doses, administered every other day, were 36.25 Gy in 5 fractions and 42.7 Gy in 7 fractions, respectively. A contrast-free computed tomography (CT) scan with a slice thickness of 1 mm was acquired for all patients, who were immobilized in the supine position with their ankles fixed through the FeetFix system (CIVCO radiotherapy, Iowa, US). Prior to the simulation CT and each treatment fraction, the rectum was emptied through a micro enema, administered in the department, and the bladder was filled by the patient drinking 500 mL of still water half an hour before the procedures. The clinical target volume (CTV) was defined as the prostate gland plus seminal vesicles, and it was expanded by 5 mm in every direction, except for the rectum interface (3 mm), to obtain the planning target volume (PTV). 

The original treatment plans were designed using a dual-arc (DA) technique with flattening filter-free (FFF) 6 MV or 10 MV photon beam energy on a VersaHD (Elekta AB, Stockholm, Sweden) linear accelerator equipped with the Agility Multileaf Collimator (MLC, 160 leaves, 5 mm thickness, up to 6.5 cm/s, MU calibration 1 MU = 1 cGy). The two arcs were almost-full arcs of 300° extension, excluding the posterior entrance to the rectum, in increments of 30°. At first glance, the DA plans were optimized involving only a 0° collimator angle, with the introduction of a 90° collimator rotation for the second arc when considered essential to fulfill or improve the dosimetric goals. The minimum target coverage criteria and the institutional protocol’s organs-at-risk (OARs) constraints, as reported in Table 1, were pursued whenever feasible. In the case of small bowel–PTV overlap or suboptimal bladder filling, the sparing of the OARs was prioritized over target coverage. The sequencing parameters utilized in the a priori Multi-Criteria Optimization for Monaco v5.51.11 (Elekta AB, Stockholm, Sweden) treatment planning system (TPS) are shown in Figure 1a. A 2 mm dose grid and a 1% statistical uncertainty per Monte Carlo calculation have been used for all DA plans.

The DA treatment plans were re-planned by two medical physicists using the same margins for the PTV but adopting a single-arc (SA) approach and incorporating new optimization parameters within the template, guided by the increased planning and clinical expertise gained over time. A photon beam energy of 10MV FFF, an arc length of 300° in increments of 20°, and a 90° collimator angle were used for all cases. The same 2 mm dose grid and a 1% statistical uncertainty per Monte Carlo calculation were applied, and the other sequencing parameters used are illustrated in Figure 1b. An SA plan optimization was developed to achieve the same aforementioned dosimetric goals (Table 1), but new cost functions and relative weights were set to reduce as much as possible the mean and intermediate doses to the OARs, especially the bladder and the rectum. 

The target and OAR dosimetric parameters of the newly optimized SA plans were evaluated and compared with the original DA plans. The relationships of rectum and bladder volumes with the rectum and bladder dose metrics were also investigated by using Pearson’s correlation coefficient and the coefficient of determination (R^2^). For the clinical assessment, one senior radiation oncologist performed a blind choice between DA and SA plans, based on dose distribution and dose-volume histograms (DVH). Each SA plan was delivered on the same Linac and measured with a perpendicular diode matrix phantom (Delta4^+^; ScandiDos, Uppsala, Sweden) to check the deliverability and the agreement between the calculated and the actual administered dose, as per institutional standards. Monitor units (MU), treatment delivery time, and gamma analysis-passing ratio (PR) with 2% (local)–2 mm criteria were recorded for each pair of plans. Student’s paired *t*-test (Stata v 9.0, StataCorp LLC, College Station, TX, USA) was used for the comparison between SA and DA plans, and a *p*-value < 0.05 was considered to be statistically significant.

## 3. Results

The prostate cancer plans included small bowel–PTV overlap (n = 3), suboptimal bladder filling (n = 3), and double prosthesis (n = 1). Median [range] CTV and PTV volumes were 52.2 cc [22.6–114.9] and 107.6 cc [53.9–199.4], respectively. Mean target coverage for the DA and SA plans, with the corresponding mean relative dose differences, are reported in Table 2. While the differences in PTV D95% and PTV D2% between the two techniques were not statistically significant (*p* = 0.058 and *p* = 0.260, respectively), PTV Dmean was significantly higher in SA plans (*p* < 0.001). The minimum target coverage objective (PTV D95% > 95%) was achieved in all but four DA plans and six SA plans, respectively. The maximum target dose constraint (PTV D1cc < 105%) was fulfilled in all but two DA plans. The mean PTV V105% value was 0.28 cc [0.00–3.45] for DA plans and 0.23 cc [0.00–0.60] for SA plans. 

Median [range] rectum and bladder volumes were 56.2 cc [28.8–86.4] and 142.7 cc [44.3–461.5], respectively. A significantly increased OAR sparing was observed in SA plans, especially in rectum and bladder mean and intermediate doses. The mean absolute dose difference was −3.7 Gy [range, −7.6–−0.6; *p* < 0.001] for rectum Dmean, −2.1 Gy [range, −7.1–1.5; *p* < 0.001] for rectum D10%, −5.1 Gy [range, −13.4–−0.1; *p* < 0.001] for rectum D20%, −5.7 Gy [range, −10.6–−0.9; *p* < 0.001] for rectum D50%, −1.2 Gy [range, −4.0–0.4; *p* < 0.001] for bladder Dmean, −1.4 Gy [range, −6.6–1.4; *p* < 0.001] for bladder D10%, and −1.9 Gy [range, −7.0–1.4; *p* < 0.001] for bladder D40%. Conversely, maximum doses to rectum and bladder were higher in SA plans, and the differences in rectum D1cc (+0.5 Gy [range, −2.0–3.2]; *p* = 0.002), rectum wall D0.035cc (+0.8 Gy [range, −0.4–2.4]; *p* < 0.001), and rectum mucosa D0.035cc (+0.6 Gy [range, −2.4–4.3]; *p* = 0.007) reached the statistically significant level. Mean values of the rectum and bladder dose metrics for the DA and SA plans, with the corresponding mean relative dose differences, are presented in Table 3. The dose to 5% of the femoral head volumes were comparable between the two techniques (Left Femur, SA: 14.5 Gy [8.8–18.0]; DA: 14.4 Gy [9.3–18.6]; *p* = 0.268; Right Femur, SA: 14.6 Gy [7.7–19.3]; DA: 14.6 Gy [10.2–20.2]; *p* = 0.476). Minor violations (<5%) of the rectum, bladder, and femoral heads dose constraints (Table 1) were, respectively, accepted in four, nine, and nine DA plans, and in two, three, and six SA plans. The only association was observed in the bladder volume that moderately correlated with the bladder Dmean (R^2^ = 0.58) and bladder D40% (R^2^ = 0.61). 

According to the physician’s evaluation, in all cases, the SA optimization technique resulted in a clinically acceptable and better treatment plan than the original one. Figure 2 shows a side-by-side comparison of the dose distribution and the DVH between DA and SA plans for one intermediate-risk prostate cancer patient.

With the use of the SA planning strategy, there was a significant reduction in the mean treatment delivery time by 22% [range, −39.7–−4.5] (*p* < 0.001). This decrease led the average delivery duration to shift from 2.1 min [range, 1.7–3.0] to 1.5 min [range, 1.3–1.9]. The differences in treatment time were reported for each patient in Figure 3. The mean MU number rose from 1819 ± 332 in the DA plans to 1967 ± 301 in the SA plans (*p* < 0.001), due to the higher plan complexity (Figure 1b). Despite the increased fluence modulation of the SA plans, the dose measurements reported an optimal agreement with dose calculations, with all PR greater than 95% for 2% (local)–2 mm criteria (SA: 98.7% [96.0–100.0]; DA: 98.0% [94.5–100.0], *p* = 0.004).

## 4. Discussion

The dosimetric quality and the treatment efficiency of single-arc (SA) versus dual-arc (DA) approaches in linac-based SBRT for localized prostate cancer were investigated by re-planning 40 VMAT plans used for treatment during 2021 at a single institution. 

Both planning modalities provided clinically acceptable treatment plans for all patients. The blind choice confirmed the improvement in the SA plans, but this was primarily influenced by the reduction in doses to the OARs rather than by changes in target coverage. One of the paramount objectives in the re-planning strategy was, indeed, to minimize the dose delivered to the rectum and bladder to the lowest achievable level while maintaining an equivalent target coverage. 

Recent data from PACE B [12] revealed that the incidence of grade (G) 2+ genitourinary (GU) and gastrointestinal (GI) toxicities in patients who underwent prostate SBRT was 23% and 10% in the acute phase and 3% and 2% in the late period, respectively. The persistence of treatment-related late toxicities, such as urinary incontinence and rectal discomfort, may significantly decrease patients’ quality of life and remains a major concern in the adoption of extreme-hypofractionated regimens. Few studies [31,32] showed a significant association between GU and GI side effects and OARs planning dose-volume parameters using normo-fractionated (1.8–2.0 Gy per fraction) prostate radiotherapy. In the context of SBRT, there are limited published data available as of today. A post-hoc analysis of patients treated within the MIRAGE trial [33] identified that an increased number of patients experienced G2+ urinary toxicity when they received higher intermediate doses in the bladder and had larger portions of the trigone exposed to elevated doses, although the trend did not reach statistical significance. However, it is reasonable to believe that optimizing the doses to the rectum and bladder would be beneficial in preventing these specific side effects. Interestingly, Kang et al. [34] found that the use of partial arcs, avoiding the posterior entrance to the rectum, resulted in a significant reduction of the rectum mean dose while significantly increasing the near-to-maximum dose of the left and right femoral heads compared to the full-arc arrangements. Similarly, in all our DA and SA plans, a partial arc arrangement was used, and it is worth noting that when optimizing for rectal and bladder sparing, the dose distribution tended to spread laterally. Therefore, a thorough optimization was also required for the femoral heads to achieve comparable values between the two techniques.

Leeman et al. [35] analyzed 23 prospective clinical trials and demonstrated that G2+ urinary toxicity in patients undergoing SBRT was mainly linked to the maximum urethral dose. In a more recent systematic review, Le Guevelou et al. [36] reported that when limiting the urethra Dmax to 90 GyEQD2 (α/β = 3 Gy), the occurrence of late G2 GU toxicity remained relatively low, and it was further decreased by reducing the urethral dose below 70 GyEQD2. In a current clinical practice, where patient catheterization or MR imaging was not employed, and the urethra could not be accurately delineated and tracked during the treatment, we adopted a urethra protection strategy. This approach involved planning for a target dose distribution that prioritized homogeneity and minimized the hotspots not only at the center of the gland, where the urethra is supposed to be, but throughout the entire target volume. The PTV D1cc restricted below 105% of the prescription dose did indeed ensure limitation of the dose to the urethra below 80–85 GyEQD2 in all our treatment plans, thus minimizing the risk of the aforementioned GU adverse effects.

Some investigators [37,38] evaluated the benefit of multi-arc VMAT plans in prostate cancer and documented their improved target homogeneity compared to single-arc plans. However, Nguyen et al. [39] stated that under the condition of unrestricted control over hardware performance parameters like dose rate, MLC leaf positioning, and gantry rotation speed, a single arc could produce the same dose distribution as multiple arcs. Likewise, Kang et al. [34] obtained very similar dose distributions for target coverage by investigating different single- and dual-arc arrangements in 18 SBRT–VMAT prostate patients. In the current study, the lack of statistically significant variations in PTV D95% and D2% between SA and DA plans supports and validates these observations.

These results were achievable due to the accumulated expertise acquired over the years of working with prostate cancer patients and using the same planning system. Sasaki et al. [40] retrospectively analyzed 148 plans over eight years using the PlanIQ™ software for learning curve evaluation and showed a trend of improvement in the quality of the plans over each successive year. Moreover, in all cases where re-planning was performed, the new treatment plan was better than the initial clinical plan administered to patients. They also observed a decrease in variability over the years compared to the earlier treatment plans. In the current study, the delivery time variability was reduced by using consistent single-arc geometry for all plans, as also shown in Figure 3. The dosimetric variability was minimized by having only two experienced medical physicists involved in the re-planning process and by using the same template for all plans. Any remaining variability is attributed to differences in patient anatomy, despite the consistent adherence to the same pre-treatment preparation regimen. 

Minor violations of the OARs dose constraint were accepted in both the DA and SA plans. In cases where the patient’s anatomy was inherently unfavorable, the urinary continence status was poor, and an improvement was not achievable in a short time, the treating physician deemed it clinically acceptable to slightly exceed certain constraints. This was especially observed for the bladder, mainly due to consistent overlap with the PTV or limited filling. Bladder volume, indeed, was moderately correlated with mean bladder doses. However, Kong et al. [41] stated that favorable bladder sparing and better compliance to protocol dose constraints were achieved during treatment if a smaller bladder was used at the time of planning. Moreover, it is important to highlight that our institutional constraints (Table 1) are quite stringent, especially for the seven-fraction regimen, which was the primary source of minor constraint violations. As a matter of fact, we used the constraints from the five-fraction regimen, recalculated using the linear quadratic model with an α/β value of 3, in addition to those from the Hypo–RT trial [9], which employed just two constraints for the rectum and one for the femoral heads.

In terms of efficiency, SA plans had shorter delivery times than DA plans. The benefit of reducing treatment time in prostate SBRT to potentially mitigate the effects of intrafraction motion has been demonstrated by several groups [14,15,16,17,18,19]. In previous studies [19,26], it has been demonstrated that the probabilities of prostate motion >3 mm and >5 mm within 5 min were 1.8% and 0.0%, respectively, and that intrafraction prostate motion had a noticeable impact on some individual fractions using 2 mm PTV margins but tended to average out when considering the cumulative effect over 4–5 treatment fractions. The use of SA VMAT plans as planning strategy to shorten the delivery time has been widely recommended as well [34,37,38,42]. In the current study, partial 10 MV FFF SA plans allowed us to keep the mean delivery time as low as 1.5 min, with all plans delivered in less than 2 min. For the setup phase, the institutional protocol consisted in the acquisition of a CBCT with fast prostate preset and a soft tissue-matching using a mask created by a 5mm expansion of the CTV, followed by manual refining in the X-ray Volume Imaging XVI software v5.0.4 (Elekta AB, Stockholm, Sweden), for a total duration of about 3 min. The short overall treatment time, along with the use of 5 mm/3 mm posterior margins, may prove to be sufficient for obtaining an adequate coverage of the target volumes and OARs sparing despite intrafraction prostate motion and setup errors. In addition, even though slightly inferior dose distributions may be justified in favor of a shorter treatment duration, our study demonstrated that SA plans can achieve comparable and even better results than DA plans. Thus, we chose to employ the SA approach for all patients with localized prostate cancer undergoing SBRT.

As far as deliverability, the rise in MUs is mainly associated with the use of different physical plan parameters in SA plans. Indeed, previous investigators [34,38] observed that MUs significantly increase with the number of arcs used in SBRT–VMAT plans and that single-arc plans led to fewer MUs than dual-arcs plans. However, those parameters were instrumental in enhancing the treatment quality and did not result in a lower PR during pre-treatment verification. This study demonstrated that SA plans showed a stronger agreement with the calculated dose, with statistically significant differences in PR values in favor of the SA technique.

This study was primarily focused on the optimization of treatment in prostate SBRT–VMAT. While it provides valuable insights into the technique’s strategies and effectiveness, long-term follow-up data are required for a comprehensive assessment of the clinical benefits of the use of SA plans. Further analyses are continuing to investigate if a significant OAR dose reduction translates into a toxicity reduction in real-world clinical prostate SBRT practice.

## 5. Conclusions

SA VMAT planning achieved clinically equivalent target coverage while significantly reducing dose to the rectum and bladder compared to DA plans, making it an attractive technique for localized prostate SBRT. Moreover, the substantial reduction in treatment delivery time decreases the probability of prostate motion beyond the applied margins. These findings suggest a potential decrease in treatment-related toxicity and improved target coverage in prostate treatments. Further investigations are warranted to assess the long-term clinical outcomes associated with this planning technique.

## Figures and Tables

**Figure 1 cancers-16-00013-f001:**
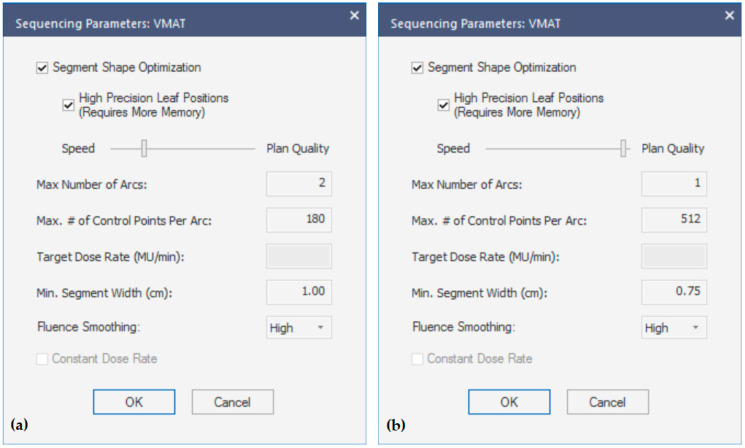
Sequencing parameters, such as number of control points per arc, segment width, and fluence-smoothing parameters, for the DA plans (**a**) and SA plans (**b**).

**Figure 2 cancers-16-00013-f002:**
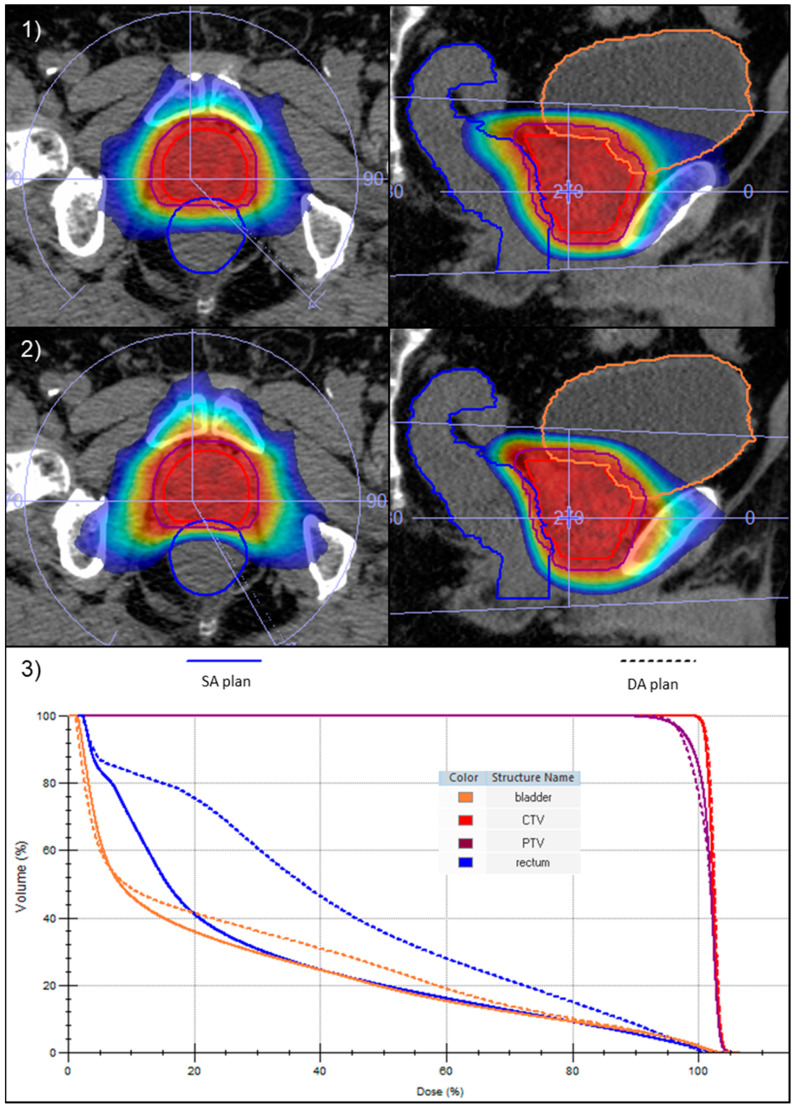
Comparison between DA (**1**) and SA (**2**) dose distributions in the axial (left) and sagittal (right) planes, with isodoses ranging from 50% to 105% of the prescribed dose. (**3**) Superposition of the DVHs of the CTV, PTV, rectum, and bladder for DA (dashed line) and SA (solid line) plans of the same patient.

**Figure 3 cancers-16-00013-f003:**
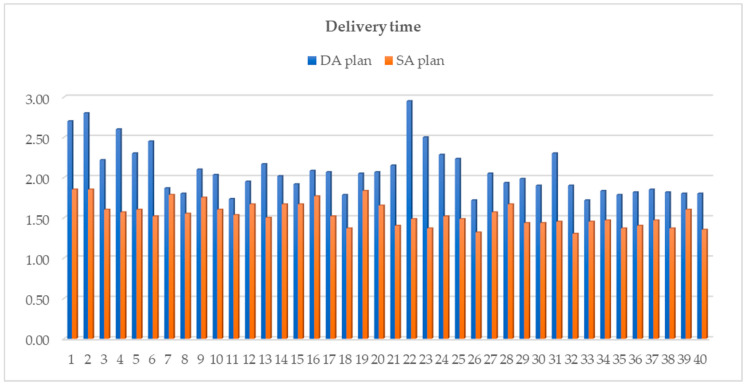
Differences in treatment delivery time for each patient between DA (blue-colored) and SA (orange-colored) plans.

**Table 1 cancers-16-00013-t001:** Treatment planning objectives for prostate SBRT in 5 and 7 fractions.

	36.25 Gy/5fx	42.7 Gy/7fx
PTV	D95% ≥ 95%	D95% ≥ 95%
	D1cc < 105%	D1cc < 105%
Rectum	D1cc < 38.06 Gy	D1cc < 44.84 Gy
	D5% < 36.25 Gy	D5% < 42.7 Gy
	D10% < 32.625 Gy	D10% < 38.43 Gy
	D20% < 29 Gy	D20% < 34.16 Gy
	D50% < 18.1 Gy	D50% < 20.3 Gy
Bladder	D1cc < 38.06 Gy	D1cc < 44.84 Gy
	D10% < 32.625 Gy	D10% < 37.3 Gy
	D40% < 18.1 Gy	D40% < 20.3 Gy
Femoral Heads	D5% < 14.5 Gy	D5% < 16.1 Gy
Small Bowel	D0.1cc < 35 Gy	D0.1cc < 40.1 Gy

**Table 2 cancers-16-00013-t002:** Mean and range of PTV dose metrics (expressed in % of the prescribed dose) for DA and SA plans, with the corresponding mean and range of the relative dose difference between the two approaches.

	Metrics	DA Plans	SA Plans	Relative Dose Difference
		Mean [Range]	Mean [Range]	Mean [Range]
PTV	D95%	96.5% [88.1–98.7]	96.7% [89.6–99.5]	+0.3% [−1.6–2.4]
	Dmean	101.1% [99.8–102.5]	101.4% [100.4–102.2]	+0.3% [−0.4–1.8]
	D2%	103.9% [102.0–105.3]	104.1% [103.3–104.6]	+0.2% [−0.7–1.7]

**Table 3 cancers-16-00013-t003:** Mean and range of rectum and bladder dose metrics (expressed in Gy) for DA and SA plans, with the corresponding mean and range of the relative dose difference between the two approaches.

	Metrics	DA Plans	SA Plans	Relative Dose Difference
		Mean [Range]	Mean [Range]	Mean [Range]
Rectum	Dmean	15.8 Gy [10.4–21.7]	12.0 Gy [8.7–17.5]	−23.2% [−41.6–−4.3]
	D1cc	38.0 Gy [33.1–43.0]	38.5 Gy [31.2–43.7]	+1.4% [−5.8–8.0]
	D5%	35.9 Gy [30.9–42.1]	35.6 Gy [26.4–42.1]	−0.9% [−14.5–8.6]
	D10%	32.7 Gy [26.3–40.3]	30.6 Gy [19.3–39.4]	−6.4% [−26.8–5.0]
	D20%	26.6 Gy [18.1–36.2]	21.4 Gy [12.5–33.1]	−19.4% [−42.4–−0.5]
	D50%	13.1 Gy [7.0–22.1]	7.5 Gy [4.0–13.4]	−42.2% [−68.7–−7.7]
Rectum wall	D0.035cc	39.9 Gy [36.1–43.9]	40.7 Gy [36.9–44.7]	+2.1% [−0.9–5.6]
Rectum mucosa	D0.035cc	37.3 Gy [33.1–42.6]	37.8 Gy [31.0–44.0]	+1.5% [−6.4–11.4]
Bladder	Dmean	13.0 Gy [5.0–22.0]	11.8 Gy [4.7–18.1]	−8.5% [−20.5–2.8]
	D1cc	40.1 Gy [36.4–43.8]	40.1 Gy [36.0–44.1]	0.0% [−3.4–3.0]
	D10%	33.1 Gy [19.0–41.6]	31.7 Gy [16.0–39.1]	−4.3% [−19.9–6.3]
	D40%	12.5 Gy [1.1–21.7]	10.7 Gy [1.3–20.7]	−12.9% [−48.2–37.2]

## Data Availability

Research data are stored in an institutional repository and can be shared upon reasonable request to the corresponding author.

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
