# Peer review of "Treatment Optimization in Linac-Based SBRT for Localized Prostate Cancer: A Single-Arc versus Dual-Arc Plan Comparison"

_cancers, 2023, doi:10.3390/cancers16010013_

Round 1

Reviewer 1 Report

Comments and Suggestions for Authors

I have read with great interest this article concerning treatment planning strategies for prostate SBRT, between dual and single arc for VMAT planning. As already highlighted by the authors, shorter treatment times for delivery are related to improved outcomes, especially for toxicity. The paper is well written and data are properly collected and presented. I have no major observations for this study. I recommend it for publication in the present version. 

Author Response

Thank you for the time and efforts you have spent in reading and reviewing our manuscript, and for the positive feedback that was really appreciated.

Reviewer 2 Report

Comments and Suggestions for Authors

The manuscript titled "Treatment Optimization in Linac-Based SBRT for Localized Prostate Cancer: A Single-Arc Versus Dual-Arc Plan Comparison" by Denis Panizza et al. is a comprehensive and well-conducted study focusing on improving the efficiency and effectiveness of stereotactic body radiation therapy (SBRT) for prostate cancer. This study is a valuable contribution to the field of oncological treatment optimization, particularly for localized prostate cancer.

  1. The study compares single-arc (SA) and dual-arc (DA) volumetric modulated arc therapy (VMAT) plans, an area that has significant implications for clinical practice in radiation oncology. The authors used a well-defined sample of 40 VMAT treatment plans, demonstrating thoroughness in the selection and re-planning process, ensuring that the study’s findings are grounded in a solid experimental framework. The manuscript not only evaluates the dosimetric quality and treatment efficiency but also includes a detailed analysis of organ at risk (OAR) sparing, treatment delivery time, and pre-treatment verification’s gamma analysis-passing ratio. This multi-faceted approach adds depth to the study's findings. The study clearly demonstrates the advantages of SA VMAT planning, particularly in reducing doses to the rectum and bladder, and reducing treatment delivery time, which is crucial in SBRT for localized prostate cancer.
  1. The authors used appropriate statistical methods, such as Student’s paired t-test, ensuring the reliability of their findings. The significance of their results is clearly stated and adds credibility to the study. The findings have immediate clinical implications, especially in terms of reducing treatment-related toxicity and improving target coverage in prostate treatments. This could potentially lead to improved patient outcomes in the long term. The manuscript rightly suggests the need for further investigations to assess long-term clinical outcomes associated with the planning technique. This shows the authors' understanding of the broader context and future applications of their research.

Overall, this manuscript is a well-structured, rigorously conducted study that provides valuable insights into treatment optimization in linac-based SBRT for localized prostate cancer. It successfully demonstrates the advantages of SA VMAT planning over DA planning, offering a meaningful contribution to the field of radiation oncology. The findings are likely to have significant impact on future clinical practice, making this study an important read for professionals in the field.

Author Response

Thank you for dedicating your time and effort to accurately read and review our manuscript. We truly appreciate your positive feedback, particularly valuing the detailed and accurate analysis you provided.